# Sub-molecular modulation of a 4f driven Kondo resonance by surface-induced asymmetry

Ben Warner[1,2], Fadi El Hallak[1,†], Nicolae Atodiresei[3], Philipp Seibt[1,2], Henning Prüser[1], Vasile Caciuc[3], Michael Waters[4], Andrew J. Fisher[1,2], Stefan Blügel[3], Joris van Slageren[5] & Cyrus F. Hirjibehedin[1,2,6]

Coupling between a magnetic impurity and an external bath can give rise to many-body quantum phenomena, including Kondo and Hund's impurity states in metals, and Yu-Shiba-Rusinov states in superconductors. While advances have been made in probing the magnetic properties of d-shell impurities on surfaces, the confinement of f orbitals makes them difficult to access directly. Here we show that a 4f driven Kondo resonance can be modulated spatially by asymmetric coupling between a metallic surface and a molecule containing a 4f-like moment. Strong hybridization of dysprosium double-decker phthalocyanine with Cu(001) induces Kondo screening of the central magnetic moment. Misalignment between the symmetry axes of the molecule and the surface induces asymmetry in the molecule's electronic structure, spatially mediating electronic access to the magnetic moment through the Kondo resonance. This work demonstrates the important role that molecular ligands have in mediating electronic and magnetic coupling and in accessing many-body quantum states.

[1] London Centre for Nanotechnology, University College London (UCL), London WC1H 0AH, UK. [2] Department of Physics & Astronomy, University College London, London WC1E 6BT, UK. [3] Peter Grünberg Institut and Institute for Advanced Simulation, Forschungszentrum Jülich and JARA, D-52425 Jülich, Germany. [4] School of Chemistry, University of Nottingham, Nottingham NG7 2RD, UK. [5] Institut für Physikalische Chemie, University of Stuttgart, 70569 Stuttgart, Germany. [6] Department of Chemistry, University College London, London WC1H 0AJ, UK. † Present address: Seagate Technology, Derry BT48 0BF, UK. Correspondence and requests for materials should be addressed to N.A. (email: n.atodiresei@fz-juelich.de) or to C.F.H. (email: c.hirjibehedin@ucl.ac.uk).

Remarkable quantum ground states can arise when magnetic moments are coupled to a conducting bath. For individual magnetic moments in metals, hopping between the orbitals that host an atomic spin and delocalized electrons can induce screening of the localized atomic moment: the Kondo effect[1]. However, even in the limit of strong charge fluctuations, it is also possible for the magnetic moment to survive if multiple orbitals are involved in the hopping, producing a Hund's impurity[2]. The situation becomes even more complex in the presence of a superconducting state, where interactions between the magnetic moment and Cooper pairs can produce spin-polarized ground states[3] that compete with the Kondo state. When coupled together in extended lattices, these impurities produce even more exotic phases, forming heavy fermion systems[1], Hund's metals[4] and Majorana bound states[5].

The atomic-scale study of individual spins hosted in d-shell metal atoms or organic radicals on metallic and superconducting surfaces has greatly increased understanding of these many-body phenomena[2,6,7]. For magnetic moments hosted in atomic f shells, such studies are more difficult because the spatial confinement of the orbitals restricts their coupling to charge transport[6,8,9]. To facilitate and even tune these interactions, molecular ligands can be utilized to mediate the coupling of the magnetic impurities to the local environment[10]. Here Lanthanide double-decker phthalocyanines (LnPc$_2$), in which a single lanthanide atom is sandwiched between two Pc rings, are ideally suited because they exhibit novel magnetic behaviour[11,12] and their high stability enables them to be studied in ultra-high vacuum conditions[13–18]. Coupling to the 4f magnetic moment in LnPc$_2$ molecules via charge transport is complicated by an additional spin arising from an unpaired electron shared between the Pc ligands[19], which has been accessed directly in scanning tunnelling microscopy (STM) studies of TbPc$_2$ on Au(111) (ref. 17). The influence of the Tb magnetic moment has been observed in nanojunction charge transport measurements[20–24], though it is difficult to determine the exact binding configuration of the molecule and, therefore, the charge transport pathway through it. More recent work has shown that it is possible to access the 4f states in NdPc$_2$ (ref. 25) directly on Cu(001) because of the strong interaction between the Pc ligand and the metal substrate.

Here we report that a Kondo resonance can be observed from a 4f-like magnetic moment in a single molecule magnet that is strongly coupled to an underlying metallic surface, and that the strength of this many-body resonance can be spatially modulated by asymmetric local intramolecular variations in the coupling between the molecule and the surface. Using a combination of STM imaging and spectroscopy experiments as well as density functional theory (DFT) modelling, we find that for dysprosium double-decker phthalocyanine (DyPc$_2$) on Cu(001) a strong hybridization between the Dy d and f orbitals enables the Dy magnetic moment to be strongly coupled to the external bath of conduction electrons, thereby inducing a 4f-driven Kondo resonance. The misalignment of the fourfold symmetric lower Pc ligand with respect to the fourfold symmetric Cu(001) surface induces an asymmetry in the electronic structure in the molecular ligands, which then spatially mediates electronic access to the 4f-like magnetic moment through the Kondo resonance. These results highlight the important role that molecular ligands have in enabling access to novel magnetic and many-body quantum states, particularly in 4f systems that are normally difficult to access electrically.

## Results

**DyPc$_2$ on Cu(001).** STM topography (see Methods) of DyPc$_2$ on Cu(001) can be seen in Fig. 1a. The top Pc ligand is observed to

sit at a mean angle of $\pm 27°$ ($\pm 2.5°$ s.d.) to the $<010>$ axis. Assuming a rotation of approximately 45° between the top and bottom ligands[18], the lower Pc ligand binds at $\mp 18°$ (Fig. 1b), which agrees with previous measurements of transition metal Pc molecules on a Cu(001) substrate[26]. The narrow distribution of angles suggests the molecule is strongly interacting with the surface, and this is further confirmed by the lack of aggregation for similar TbPc$_2$ molecules[27]. As previously observed[13,14,16,18,25], the eight-lobe structure of the Pc$_2$ molecule observed in STM imaging is due to the electronic structure of the top ligand; given the configuration of the molecule, it is not possible to tunnel directly into, and therefore image, the lower Pc ligand.

Figure 1c shows high voltage d$I$/d$V$ scanning tunnelling spectroscopy measurements acquired with the STM tip placed over the centre and the ligands of the DyPc$_2$. Although these spectra appear similar to those observed for NdPc$_2$ on Cu(001)

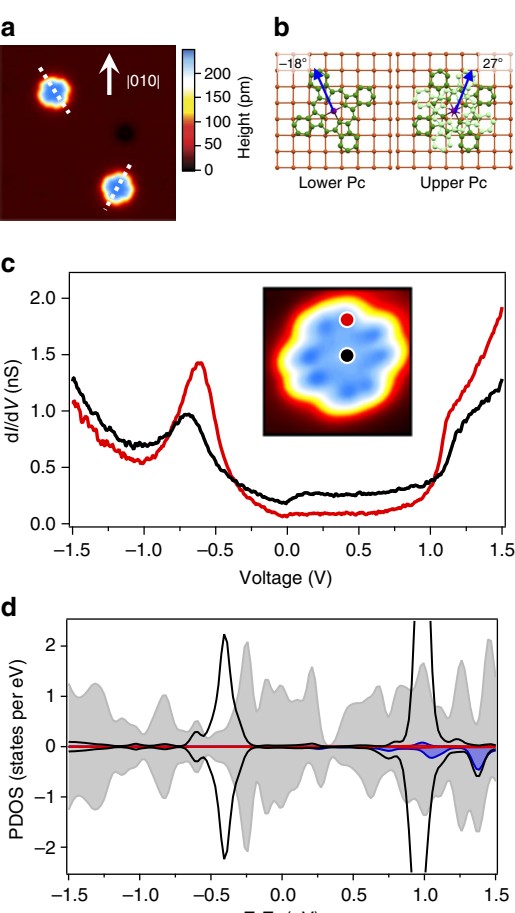

**Figure 1 | DyPc$_2$ on Cu(001).** (a) STM topographic image (18.6 × 18.6 nm; $V_{set} = -0.1$ V, $I_{set} = 0.1$ nA) of DyPc$_2$ molecules, which are observed to bind at two angles, as marked with a dotted white line. (b) Schematic of the binding of molecules to the surface, showing the lower (dark green) and upper (light green) Pc ligands, with the centre of the molecule placed over the hollow site in the Cu(001) surface[25]; the central Dy atom is shown. Pc molecules have been observed to bind at $\pm 18°$. For an assumed rotation of 45°, the top Pc ring will appear at $\mp 27°$, which is approximately the observed angle of the molecules. (c) High-voltage spectroscopy ($V_{set} = -1.5$ V, $I_{set} = 0.1$ nA) acquired over the ligand (black) and the centre (red) of a DyPc$_2$ molecule (positions shown in inset on an STM topographic image, 3.5 × 3.5 nm, $V_{set} = -0.2$ V, $I_{set} = 0.1$ nA). (d) Calculated partial density of states (PDOS) of π states in the top (black) and bottom ligand (grey), and Dy d- (red) and f-states (dark blue). The lower ligand is strongly hybridized with the surface and also weakly hybridized to the top ring.

(ref. 25), they show some clear differences. For example, the prominent peak in the DyPc$_2$ local density of states at approximately $-0.7$ V and the step at $+1.0$ V are closer to the Fermi energy by $\sim 0.2$ V. Furthermore, an enhanced local density of states is observed below $-1.0$ V and a new spectroscopic feature is observed close to the Fermi energy.

To understand the spectrum in more detail, we carried out DFT calculations (see Methods) of DyPc$_2$ on Cu(001). As shown in the PDOS plots in Fig. 1d, the bottom Pc ligand is strongly hybridized with the Cu(001) surface while the upper Pc ligand is only weakly coupled. Note that features associated with empty f-orbital states (positive energy) are seen in the same approximate energy range as the states of the upper Pc ligand, while none are observed between $-1.5$ eV and the Fermi energy. We, therefore, assign the prominent spectroscopic features at approximately $-0.7$ V and $+1.0$ V to states of the upper Pc ligand. The calculations also indicate that, as is seen for NdPc$_2$ on Cu(001) (ref. 25), the top ligand states are shifted in energy by the electric field applied between the STM tip and the metallic surface. In our spectroscopic measurements, these features are only observed when their energy is pinned by other nearby states.

To investigate the new low-bias feature further, higher resolution d$I$/d$V$ measurements were taken at low bias over both the centre of the molecule and on the two sides of the outer ligands (Fig. 2a). In all cases, a Fano line shape[28] near the Fermi energy is observed in the spectra, which can be fitted to the form:

$$\frac{dI}{dV} = C + \frac{A}{(1+q^2)}\frac{(q+\epsilon^2)}{(1+\epsilon)^2}; \epsilon = \frac{eV - E_k}{\Gamma} \quad (1)$$

where $e$ is the magnitude of the electron charge; $C$ is a constant offset of the spectra representing background conductance through other channels; $A$ is the amplitude of the resonance; $q$ is the ratio of tunnelling into the resonance and into the continuum[6]; $E_k$ is the offset from the Fermi energy (zero bias); $\Gamma$ is the resonance width; and $V$ is the bias voltage. Such a feature is created when two coherent tunnelling paths interfere with each other, where one path is into a sharp resonance and the second is into a broad continuum[28]. The line width $\Gamma$ is related to the strength of the interaction between the resonance and continuum, and the amplitude of the Fano line shape $A$ is determined by the continuum of the system.

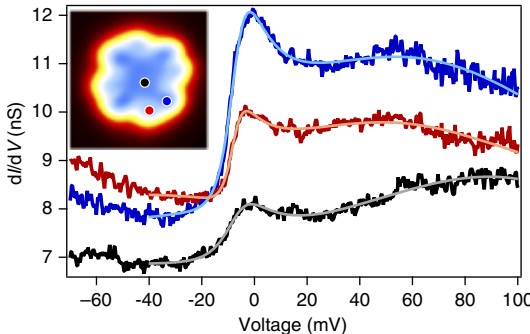

**Figure 2 | Spatial variation of Fano line shape.** d$I$/d$V$ spectroscopy ($V_{set} = -70$ mV, $I_{set} = 0.5$ nA) acquired over the centre (black) and two sides (red and blue) of the top ligand of the DyPc$_2$ molecule, vertically offset for clarity. A clear difference in amplitude is observed on alternating sides of the ligands. A Fano feature is observed near 0 V, along with a broad peak centred at $\sim 50$–60 mV on the upper Pc ligand and at $\sim 90$ mV at the centre of the molecule. Solid lines show a fit with a Fano line shape and a Gaussian. Inset shows a constant current topographic image ($V_{set} = 0.1$ V, $I_{set} = 0.1$ nA) of DyPc$_2$; the position at which each spectrum shown has been acquired is marked in the corresponding colour.

In STM studies of magnetic systems on metallic surfaces, such a sharp resonance near the Fermi energy often indicates Kondo screening[29–31]. In addition, we also observe a broad peak at more positive voltages and, therefore, include an additional Gaussian component in the fits; such additional resonances have been observed previously on other metal-doped Pc molecules on metal surface and result from a many-body orbital effect[31].

**Kondo effect from a 4$f$-like magnetic moment.** For the magnetic properties of the molecule on the Cu(001) surface, the DFT calculations indicate that the Dy atom has a large spin moment originating from the 4$f$-like atomic states; spin-orbit effects are not included in our calculations so the total moment cannot be determined. This is in contrast to some Pc molecules on surfaces that host 3$d$ metal atoms, such as CoPc on Au(111) (ref. 10), where the transition metal spin is quenched by the strong interaction with the substrate. Furthermore, unlike the case for the isolated DyPc$_2$, the Pc ligands no longer host an unpaired electron due to the charge rearrangement caused by the strong hybridization at the molecule-surface interface; similar behaviour is observed for NdPc$_2$ on Cu(100) (ref. 25).

Since the only surviving magnetic moment in the molecule is the 4$f$-like moment, it is natural to interpret the feature near zero bias as a Kondo resonance arising from the coupling of the 4$f$-like moment to the conduction electrons in the underlying Cu(001). This is in contrast to results from measurements for both TbPc$_2$ and YPc$_2$ on Au(111), in which a Kondo effect has been observed and is derived from the delocalized electron on the top Pc ligand. Confirmation of this is demonstrated by the facts that in these systems, which contain a spin in the upper Pc ligand, (i) tunnelling predominantly occurs directly into the Kondo resonance (that is, $q \gg 1$) and (ii) a Kondo resonance is observed only on the ligands and is absent at the centre of the molecule. In contrast, for DyPc$_2$ on Cu(001), we observe a Fano line shape with roughly equal tunnelling into both the Kondo resonance and the continuum ($q \sim 1$), and the resonance is observed when tunnelling both at the centre of the molecule and over the rest of the upper Pc ligand. As seen in Supplementary Fig. 1, some of the ligand states are directly coupled to the Dy atom while others are not. The value of $q \sim 1$ for the Fano resonance therefore suggests that the tunnelling electron passes roughly equally through each of these two kinds of ligand states as it tunnels between the tip and the continuum formed by the hybridization of the substrate and the lower Pc ligand (Supplementary Fig. 2).

Our calculations show that, in contrast to early series lanthanide Pc$_2$ molecules where two 6$s$ and one 4$f$ electron are formally donated to the Pc rings[25], for a Dy atom only two 6$s$ electrons are transferred to the Pc. More precisely, the total charge of $\sim 10$ electrons found in an atomic sphere around Dy implies that for the DyPc$_2$ molecule the formal oxidation state of Dy atom is $2+$ (see Supplementary Discussion and Supplementary Figs 1–5 for a more detailed discussion regarding the bonding that takes place in the DyPc$_2$ molecule). Additionally, projecting the total charge density within a sphere around the Dy atom onto the atomic-like orbitals leads to the following quantities: 0.16 in $s$, 0.07 in $p$, 0.72 in $d$ and 9.21 in $f$ channels, respectively. The partial occupancies of the $s$, $d$ and $f$ channels reveal that the interaction between the confined $f$-shell states and the ligands is mediated through atomic hybrid orbitals with mixed $d$ and $f$ character. This is in sharp contrast to 4$f$ atoms adsorbed directly on metal surfaces, where the Dy atomic $d$ and $f$ states do not mix to form atomic hybrid orbitals[8]. One important difference between these two cases is that the local chemical environment of the Dy atom is modified by the upper and lower Pc ligands, which allows the

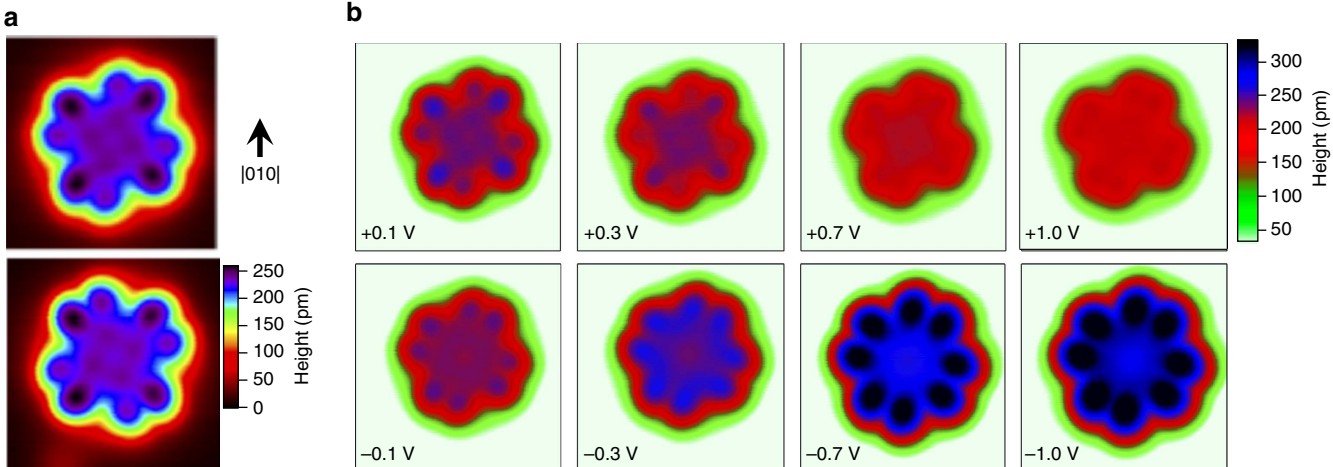

**Figure 3 | Electronic asymmetry in images of DyPc₂.** (**a**) Topographic image (4.5 × 4.5 nm, $V_{set} = 0.1$ V, $I_{set} = 0.1$ nA) of DyPc₂ in two mirror symmetric configurations. A clear asymmetry is observed, which obeys the same mirror symmetry as the underlying binding. This suggests that it is related to the relative orientation of the fourfold symmetric molecule on the fourfold symmetric surface. (**b**) Topographic images (4.5 × 4.5 nm, $I_{set} = 0.1$ nA) of DyPc₂ on Cu(001) at various bias voltages. The ligand state of the top Pc ring is observed at − 0.7 V (ref. 14). For small-positive biases, a clear asymmetry can be seen in the ligands that persists at larger-positive voltages but is absent at negative voltages.

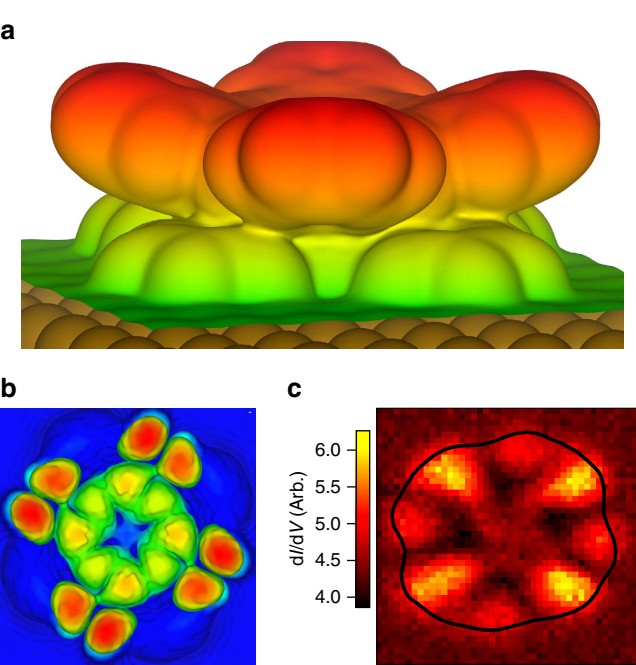

**Figure 4 | Electronic asymmetry of due to surface interaction.**
(**a**) Isosurface of the total charge distribution of DyPc₂/Cu(001) system. A coupling between the two Pc rings is observed only on one side of the arm. (**b**) Simulated STM image from − 10 to 10 mV. The asymmetry is clearly observed in the ligand states. (**c**) dI/dV slice at − 3 mV ($V_{set} = 70$ mV, $I_{set} = 0.4$ nA), with the topographic outline of the molecule shown (black line). The close match between theoretical calculation and the dI/dV slice demonstrates the continuum of the Fano lineshape is due to the ligand states, and the 4f Kondo is modulated through this. The outline of the molecule from the simultaneously taken topographic image is shown in black.

atomic *d* and *f* states to mix despite the fact that this is forbidden for isolated atoms.

Furthermore, the magnetizations of each of the atomic like channels are $− 0.004 \, \mu_B$ (*s*), $+ 0.011 \, \mu_B$ (*p*), $− 0.025 \, \mu_B$ (*d*), and $− 5.018 \, \mu_B$ (*f*); this shows that the magnetic moment of the Dy atom originates predominantly from the 4f atomic-like orbitals.

We note that a common feature of the electronic structure of the DyPc₂ molecule in gas phase as well as on the Cu(001) substrate is (i) an onsite atomic *d-f* hybridization leading to atomic hybrid orbitals with both *d-* and *f-* character that (ii) subsequently significantly overlap with the molecular electronic states of the two Pc ligands.

Given that the Dy moment in DyPc₂ has a large angular momentum $J = 15/2$ (ref. 11), it is somewhat surprising that strong Kondo screening is observed since Kondo interactions couple degenerate states with $\Delta m_j = \pm 1$. However, although DyPc₂ has a bistable ground state doublet of majority $J_z = \pm 13/2$ (ref. 32), the first set of excited states has $J_z = \pm 11/2$ and is relatively close (only a few meV away) in energy[11,32]. Because our *ab inito* calculations show a strong hybridization of the Dy to the Cu(001) via the lower Pc ligand, these Dy doublets are likely to couple due to the level broadening. This results in a ground state doublet that does have the appropriate symmetry for Kondo screening.

We note that $\Gamma$ does not vary significantly across the molecule, with an average value of $8.2 \pm 1.5$ mV. This gives a Kondo temperature of 2 $\Gamma/k_B = 33.2$ K, where $k_B$ is the Boltzmann constant. This is appreciably higher than the Kondo temperature measured for a 4f state of a single atom or cluster[6]. However, this is close to that observed for broken TbPc₂ (ref. 14); in addition, magnetic molecules have been observed to have significantly higher Kondo temperatures than single atoms[10].

**Spatial asymmetry of the Kondo effect.** As seen in Fig. 2, a stark difference in the amplitude *A* of the Fano lineshape is observed between the two lobes of the molecule. In contrast, the other parameters of the lineshape show relatively little variation across the molecule. By spatially mapping *A*, it is possible to consider the physical origins of the continuum, which in this case is composed of the ligand states of the molecule. This electronic asymmetry is also observed in the apparent height of the lobes of outer part of the ligands of the molecule for STM images taken at + 0.1 V, as shown in Fig. 3a. The mirror symmetric version of this asymmetry is observed on the molecule bound to the surface at the mirror-symmetric binding angle, suggesting that the asymmetry results from an interaction with the surface.

As shown in Fig. 3b, this asymmetric appearance of the DyPc₂ molecule is only observed in STM imaging in a specific range of

applied bias voltage, being strongest at low-positive bias and diminishing as the bias voltage is increased; at all measured negative voltages, the molecule's lobes are symmetric in apparent height. If the asymmetry observed were the result of a conformation change in the molecule[18], then we would expect the molecule to be asymmetric when imaged at both positive and negative bias voltages. That we observe the molecule is symmetric when imaging at negative bias, and that the asymmetry is not strongly affected by the height of the tip, suggests that the asymmetry is not the result of a tip-induced physical deformation of the upper Pc ligand of the molecule but rather arises because of electronic interactions with the substrate[33].

An additional manifestation of the asymmetric coupling between the $DyPc_2$ molecule and the Cu(001) substrate is the small (few degrees) rotation of the upper Pc ligand with respect to the lower Pc ligand, as seen in the DFT calculations (Fig. 1b) as compared with gas-phase $DyPc_2$ geometry in which the two ligands are 45° apart. The upper Pc ligand is too far ($\sim 6$ Å) above the copper substrate to interact with it directly. Therefore, it is natural to conclude that this arises from van der Waals interactions. Figure 4a shows a plot of an isosurface corresponding to the total charge distribution. Here a clear asymmetry is also observed between the arms of the top ring, with one side showing coupling to the lower ring, which matches very well the asymmetry observed experimentally. Note that a similar electronic asymmetry has also been observed for single-decker Pc molecules asymmetrically adsorbed on fourfold symmetric surfaces[33,34].

To further examine the electronic asymmetry, as shown in Fig. 4b, we simulated a topographic STM image for a 20 mV interval around the Fermi energy ($-10$ to $10$ mV), to approximately match the continuum sampled by the Fano lineshape. The agreement to Fig. 4c, which shows the spatial variation of $dI/dV$ at $-3$ mV (that is, near the peak in the Fano line shape), demonstrates that the coupling to the surface is creating the modulation in the continuum of the Fano lineshape and that through this we are able to observe the influence of the ligand state at an energy that would otherwise not be accessible in STM measurements. As seen in Supplementary Fig. 4, this asymmetry may arise from the tails of molecular states that lie just above the Fermi energy.

Furthermore, because the $q$ factor shows little variation across the molecule, the ratio of tunnelling into the continuum and the $4f$-like resonance is approximately constant. In contrast, the DOS of the molecule varies spatially, meaning that the total flow of current varies with tip position. Therefore, by injecting electrons at a constant height above the molecule but at different locations, it is possible to vary the amount of current going through the $4f$ Kondo state.

## Discussion

In summary, we have observed a Kondo effect due to a $4f$-like magnetic moment in a $DyPc_2$ molecule chemisorbed on Cu(001), and access to this Kondo state can be mediated with sub-molecular resolution. The Kondo resonance results from the interaction of the Dy magnetic moment with a spatially asymmetric continuum formed by strong hybridization of the metal substrate and the ligand states of the molecule. This work demonstrates that despite the close confinement of the magnetic moment in late $4f$ lanthanides, it can be directly accessed in electrical transport via controlled electronic coupling of the molecule to a metallic substrate, and that this coupling can be spatially modulated at the sub-molecular scale. This opens possibilities for using surface-molecule hybridization to access localized electronic states derived from $4f$-like orbitals.

Additionally, coupling mediated by functionalized molecular ligands could be used as a method of mediating access to many-body lattice phenomena, such as the creation of artificially constructed heavy fermion systems.

## Methods

**STM and spectroscopy.** Experiments were carried out in ultra-high vacuum using an Omicron Cryogenic STM operating at $\sim 2.5$ K (ref. 35) and an Oxford Instruments STM operating at $\sim 8$ K. Constant current topographic images and spectroscopic data were acquired with an initial set point current $I_{set}$ and a set point voltage $V_{set}$; spectra were acquired using a lock-in technique with the addition of a modulation voltage of $\sim 0.1$ mV (Figs 2 and 4c) or $\sim 3$ mV (Fig. 1c). The $dI/dV$ slice (Fig. 4c) is extracted from spectroscopic measurements obtained over an evenly spaced array of points. No significant variation in the measurements is observed with different tips.

$DyPc_2$ molecules were sublimed at $\sim 350$ °C onto a room temperature Cu(001) crystal (Matek) that had been cleaned by multiple cycles of Ar sputtering and followed by annealing to 500 °C. The molecules are normally (242 out of 253) absorbed intact on the surface. To isolate the molecules from possible interactions with each other, we used a low coverage ($\sim 2 \times 10^{-4}$–0.05 molecules nm$^{-2}$); however, on very rare occasions, as shown in Fig. 1a, molecules were found within a few nm of each other. In both our experimental setups, molecule sublimation occurred in a side chamber of the ultra-high vacuum (UHV) system with no pressure gauge. The pressure in the neighbouring chamber, which was open to the side chamber, was below $2.0 \times 10^{-9}$ mbar during the sublimation process.

**Density functional theory.** Spin-polarized, first-principles total-energy calculations have been carried out in the framework of the DFT[36] in the Kohn–Sham formulation by using the projector augmented wave method[37] as implemented in the Vienna *ab initio* simulation package code[38,39]. In our study, we used the Perdew–Burke–Ernzerhof exchange-correlation energy functional[40] and the plane-wave basis set includes all plane waves up to a kinetic energy of 500 eV. To account properly for the orbital dependence of the Coulomb and exchange interactions of the Dy $4f$-states, we employed the generalized gradient approximation with the inclusion of a Hubbard U correction (GGA+U)[41]. The Hubbard parameter for the $f$-states was set to $U_{eff} = 6$ eV to reproduce the experimental $dI/dV$ features, see main text. The $DyPc_2$-Cu(001) system was modelled within the supercell approach and contains five atomic Cu layers with the adsorbed molecule on one side of the slab. The ground-state adsorption geometry was obtained by including van der Waals interactions at a semi-empirical level[42] and by relaxing the uppermost two Cu layers and the molecular degrees of freedom until the atomic forces were converged to less 0.001 eVÅ$^{-1}$.

**Data availability.** Experimental data that support the findings of this paper are available online at DOI 10.6084/m9.figshare.3383038 (ref. 43). Additional information on the experimental data as well as details about the calculations can be obtained by contacting the authors.

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

## Acknowledgements

We are grateful for valuable discussions with Mathias Bode, Matteo Mannini, Jose Ignacio Pascual, Jascha Repp and Roberta Sessoli. B.W., F.E.H., H.P., A.J.F. and C.F.H. acknowledge financial support from the EPSRC (EP/H002367/1 and EP/D063604/1) and the Leverhulme Trust [RPG-2012-754]; M.W. and J.v.S. acknowledge support from University of Nottingham [NRF 4315]. Computations were performed under the auspices of GCS at the high performance computer JUQUEEN operated by the Jülich Supercomputing Centre (JSC) at the Forschungszentrum Jülich. N.A. and V.C. gratefully acknowledge financial support from the Volkswagen-Stiftung through the 'Optically Controlled Spin Logic' project.

## Author contributions

B.W., F.E.H., J.v.S. and C.F.H. conceived of the project; M.W. synthesized the molecules; B.W., F.E.H. and H.P. performed the experiments; B.W., F.E.H., P.S. and H.P. analysed the results; N.A. and V.C. performed the DFT calculations; all authors discussed the results and contributed to the writing of the paper.

## Additional information

**Competing financial interests:** The authors declare no competing financial interests.

