## [Peer review file · Nature Communications]

Reviewers' comments:

Reviewer #1 (Remarks to the Author):

This manuscript describes the observation of a Kondo resonance in a double decker class molecule, DyPc2, adsorbed on Cu(001) surface using low temperature scanning tunneling microscopy and spectroscopy. This observation is supported by density functional theory calculations that explain the origin of the observed Kondo effect. Although Kondo effect has been observed in many different systems and molecules on surfaces, the novelty of the current finding lies on the fact that the observed Kondo resonance is originated from a 4f-like state, which is induced by a strong hybridization between the ligands of the molecule with the substrate. Because of the hybridization, the amplitudes of the Kondo resonance are different at the caged Dy atom at the center, the upper Pc and lower Pc. The experimental images are beautiful and DFT theory is absolutely essential here to explain the electronic structure, and the asymmetry. The discovery of spin-electron interactions with 4-f like magnetic moment of the molecule that can be manipulated with the organic ligand is exciting. It is also interest for many readers. In principle, this manuscript deserves to be considered publication in Nature Communications however the following minor points need to be addressed first:

1). Fig. 2a STM image at inset: Is it a standard STM image or a dI/dV map? This needs to be clarified. If this is an STM image, then it is taken at negative bias (-70 mV) and it clearly shows one ligand is stronger (blue) in contrast. In bias dependent images shown in Fig. 3, all four images taken at negative biases show similar intensity for both upper and lower Pc. Is it because of the tip effect? Then the authors should explain how reliable and reproducible for the observed Kondo features?

2). Again in Fig. 2a: The authors should clearly label the colored dI/dV curves, red, blue and black as their origins such as upper Pc, lower Pc etc. The location of Dy at the center is clear but the upper and lower Pc location should be clarified.

3). In page 4 last paragraph, it is mentioned that the bottom Pc is strongly coupled while the upper Pc is weakly coupled to the surface. But if the blue dI/dV curve in Fig. 2a corresponds to the upper ligand, the authors need to explain why the upper Pc has stronger Kondo amplitude than the lower Pc although it is weakly coupled to the substrate.

4). Fig. 2 Caption: According to the dI/dV curves shown in Fig.2, the broad peaks at positive bias appear ~ 54 mV for red and blue curves in Fig. 2, not between 40-50 mV. The broad peak for the black curve is shifted even more positive value. These need to be corrected.

Reviewer #2 (Remarks to the Author):

'Sub-molecular modulation of a 4f driven Kondo resonance ...'
by Warner et al.

This paper describes the detection of 4f included Kondo resonance for the double-decker lanthanide , phthalocyanine (Pc) complex DyPc2. The target molecule shows a single molecule magnet behavior and attracts many attentions. Even though the magnetic behavior of the 4f orbital is the center of the interests and the STM measurement can detect spin behavior by an observation of the Kondo resonance, it is difficult to obtain the 4f spin induced Kondo resonance. The scope of the paper is timely. Although experiments and theoretical analysis have been done in a robust manner, the referee would like to list the following issues to be revised before

publication.

a) The spin which forms the Kondo resonance:

This oxidation state of DyPc2 in vacuum is calculated to be 2+, which gives no magnetic moment for the ligand pi state.

The ligand has no spin and does not contribute to the formation of the Kondo resonance, which shows a clear difference from the case of TbPc2. This situation, however, shows almost no change after adsorption on Cu(111) surface.

For various MPc and MPc2 molecules, the charge transfer from the substrate to the molecule is reported, which makes the spin on/off in the molecule. For example, even for the weak interacting CoPc on Au(111), the charge transfer from Au to Co state makes the spin of the Co disappear and no Kondo state observed for CoPc/Au(111). For TbPc2/Au(111), Vitali et al calculated the electronic structure and shows that the unpaired pi radical in the ligand for the molecule in the vacuum is filled if it is adsorbed on Cu surface. It is natural because Cu surface provides stronger bonding to a molecule than Au surface does. Since DyPc2 molecule is adsorbed on Cu surface with the same flat lying configuration, we expect similar charge transfer which might change the spin state of the ligand. The authors should state a comparison with other cases and provide the mechanism for the differences.

b) The shape of Kondo resonance

By analyzing the zero-bias peak with the Fano resonance function, the authors deduces $q \sim 1$ which suggest the tunneling electron is injected into both the Kondo resonance and the continuum. Actually the shape of the peak is similar to the Kondo resonance reported for a 3d magnetic metal atom deposited on Au(111) surface.

This q parameter suggests a strong coupling between the orbital, which is responsible for the Kondo resonance, and the substrate. It is quite reasonable for the case of 3d magnetic metal atom case. However, the 4f atom of DyPc2 is not directly attached to the surface. In addition, even the Kondo resonance is caused by its hybridization with d orbital of the Dy atom, the referee cannot consider a mechanism which makes the coupling with the substrate much stronger than that of the TbPc2, the latter of which shows a q corresponding to a weak coupling with the substrate. The authors should provide more detail analysis for the origin of the parameter q to be ~ 1 .

c) Related to issue b). The Dy atom is, in a sense, shielded by the top Pc ligand. The STM tip cannot access physically to the Dy atom, and it is assumed that the tip position is regulated largely by the DOS of the Pc ligand and not by the Dy orbitals. The tunneling electrons are injected first into the orbital of the ligand. This is totally different from the case of 3d magnetic atom on non-magnetic metal surface case. The authors should discuss how the Kondo resonance is probed with such a tunneling path.

d) comparison with bare Ln atom adsorbate

Although the authors discuss the difficulty of the detection of 4f induced Kondo resonance, there is a recent report like following. More updated description for the detection of 4f Kondo resonance should be provided.

'Absence of a spin-signature from a single Ho adatom as probed by spin-sensitive tunneling' by Steinbrecher et al. DOI: 10.1038/ncomms10454

e) Flexible configuration change of molecule; asymmetric electronic structure;

Asymmetric electronic configuration is discussed as a mechanism of the hybridization of 4f orbital with other orbitals.

However, the conformation change of the molecule can be seen for many molecule with adsorption on the surface. The authors should provide the evidence that the asymmetric STM image is not induced from the conformation change of the molecule upon adsorption.

Reviewer #3 (Remarks to the Author):

The authors present evidence of a Kondo resonance arising from the coupling between the f-states of a single-molecular magnet and the metallic states of the supporting surface. The work seems to me very carefully performed, with high scientific rigour, well inserted into the current literature, and, quite importantly, reported in a way that clearly emphasises the new physics displayed by the specific system under analysis. In my opinion this work is a valuable addition to the scientific literature of surface-supported single-molecular magnets, and would recommend publication after the authors considered the following recommendations.

The work reports two main conclusions: a) the closely confined magnetic moment carried by the 4f states can be accessed through its electronic coupling with the metallic substrate; and b) this coupling can be controlled at the molecular scale by varying the symmetry and orientation of the interacting molecule. I find that the first conclusion is very well documented, reported, and supported by clear figures. The second conclusion is in my opinion not equally well explained. Its description is limited to the last two paragraphs of the result section, that are in my opinion not fully convincing and should be expanded so as to better support conclusion b. Incidentally, the role of van der Waals interactions into this effect, i.e. change the relative rotation of one Pc ligand with respect to the other, is not evident nor obvious to me.

The authors support and explain the experimental STS measurements by high-quality DFT calculations. These reveals the hybridisation of the Dy-f states with the on-site s and d states. This conclusion is at the basis of the interpretation of the results. My question is on the specific charge population analysis used to partition the charge density and to obtain that result. This partition is non unique and there are several possibilities to do so. Projecting the charge in within a sphere of a given radius on the Dy atomic orbitals is the method used in this work. How much the reported hybridisation arises from the ligand charge density entering into the projection sphere? How would the result change as a function of the sphere radius or by employing a different charge population analysis? The authors could strengthen their conclusion by demonstrating that the reported hybridisation is independent on the chosen charge analysis.

Reviewer report and response

Nature Communications manuscript NCOMMS-16-02484-T

Reviewer #1 (Remarks to the Author):

This manuscript describes the observation of a Kondo resonance in a double decker class molecule, DyPc2, adsorbed on Cu(001) surface using low temperature scanning tunneling microscopy and spectroscopy. This observation is supported by density functional theory calculations that explain the origin of the observed Kondo effect. Although Kondo effect has been observed in many different systems and molecules on surfaces, the novelty of the current finding lies on the fact that the observed Kondo resonance is originated from a 4f-like state, which is induced by a strong hybridization between the ligands of the molecule with the substrate. Because of the hybridization, the amplitudes of the Kondo resonance are different at the caged Dy atom at the center, the upper Pc and lower Pc. The experimental images are beautiful and DFT theory is absolutely essential here to explain the electronic structure, and the asymmetry. The discovery of spin-electron interactions with 4-f like magnetic moment of the molecule that can be manipulated with the organic ligand is exciting. It is also interest for many readers. In principle, this manuscript deserves to be considered publication in Nature Communications however the following minor points need to be addressed first:

1). Fig. 2a STM image at inset: Is it a standard STM image or a dI/dV map? This needs to be clarified. If this is an STM image, then it is taken at negative bias (-70 mV) and it clearly shows one ligand is stronger (blue) in contrast. In bias dependent images shown in Fig. 3, all four images taken at negative biases show similar intensity for both upper and lower Pc. Is it because of the tip effect? Then the authors should explain how reliable and reproducible for the observed Kondo features?

Because of the configuration of the molecule on the surface, electrons are only able to tunnel between the STM tip and the top Pc ligand; it is not possible to tunnel directly into, and therefore image, the lower Pc ligand. To make this point clearly and to avoid any unintended confusion, we have added a comment to the first paragraph on p. 3.

The STM image in the inset of Fig. 2a is a constant current topographic image and not a dI/dV map. To make this point clearly and to avoid any unintended confusion, we have updated the caption for Fig. 2 to indicate this. Furthermore, this image is obtained at a voltage of +0.1 V and is therefore consistent with the corresponding image shown in Fig. 3b. To clarify the voltage at which this (and all other) STM images were obtained, we have updated the figure captions to place this text closer to the relevant part of the sentence rather than at the end.

No significant variation in the measurements is observed with different tips. We have added a comment in the Methods section to indicate this.

2). Again in Fig. 2a: The authors should clearly label the colored dI/dV curves, red, blue and black as their origins such as upper Pc, lower Pc etc. The location of Dy at the center is clear but the upper and lower Pc location should be clarified.

We have updated the caption for Fig. 2 to indicate that the black spectrum is obtained over the centre of the DyPc2 molecule while the red and blue spectra are obtained on two sides of the top ligand.

3). In page 4 last paragraph, it is mentioned that the bottom Pc is strongly coupled while the upper Pc is weakly coupled to the surface. But if the blue dI/dV curve in Fig. 2a corresponds to the upper ligand, the authors need to explain why the upper Pc has stronger Kondo amplitude than the lower Pc although it is weakly coupled to the substrate.

As described in the previous responses, the spectra in Fig. 2 are obtained on two sides of the upper Pc ligand. The changes discussed above should clarify this point.

4). Fig. 2 Caption: According to the dI/dV curves shown in Fig.2, the broad peaks at positive bias appear ~ 54 mV for red and blue curves in Fig. 2, not between 40-50 mV. The broad peak for the black curve is shifted even more positive value. These need to be corrected.

We thank the referee for pointing out this mistake in the caption and have corrected it accordingly.

Reviewer #2 (Remarks to the Author):

'Sub-molecular modulation of a 4f driven Kondo resonance ...'
by Warner et al.

This paper describes the detection of 4f induced Kondo resonance for the double-decker lanthanide, phthalocyanine (Pc) complex DyPc₂. The target molecule shows a single molecule magnet behavior and attracts many attentions. Even though the magnetic behavior of the 4f orbital is the center of the interests and the STM measurement can detect spin behavior by an observation of the Kondo resonance, it is difficult to obtain the 4f spin induced Kondo resonance. The scope of the paper is timely. Although experiments and theoretical analysis have been done in a robust manner, the referee would like to list the following issues to be revised before publication.

a) The spin which forms the Kondo resonance:

This oxidation state of DyPc₂ in vacuum is calculated to be 2+, which gives no magnetic moment for the ligand pi state.

The ligand has no spin and does not contribute to the formation of the Kondo resonance, which shows a clear difference from the case of TbPc₂. This situation, however, shows almost no change after adsorption on Cu(111) surface.

For various MPc and MPc₂ molecules, the charge transfer from the substrate to the molecule is reported, which makes the spin on/off in the molecule. For example, even for the weak interacting CoPc on Au(111), the charge transfer from Au to Co state makes the spin of the Co disappeared and no Kondo state observed for CoPc/Au(111). For TbPc₂/Au(111), Vitali et al calculated the electronic structure and shows that the unpaired pi radical in the ligand for the molecule in the vacuum is filled if it is adsorbed on Cu surface. It is natural because Cu surface provides stronger bonding to a molecule than Au surface does. Since DyPc₂ molecule is adsorbed on Cu surface with the same flat lying configuration, we expect similar charge transfer which might change the spin state of the ligand. The authors should state a comparison with other cases and provide the mechanism for the differences.

Although the oxidation state of the DyPc₂ in vacuum is 2+ from our DFT calculations, there is one unpaired electron in the spin-up channel. This gives the ligand pi state for the isolated molecule a magnetic moment. When the DyPc₂ is adsorbed on Cu(001), the strong hybridisation with between the lower Pc ligand and the surface shifts the orbitals for the Pc ligands such that spin-up and spin-down channel are equally occupied. Similar behaviour is observed by Fahrenndorf et al. [Ref. 25] for for the ligand spin of NdPc₂ on Cu(001). This quenches the ligand spin, leaving the only magnetic moment on the molecule localised on the Dy site. This is in contrast to case for CoPc on Au(111), where adsorption of the molecule on the surface quenches the Co d-spin.

To explain this more clearly and to provide a comparison with other work, we have added text to the last paragraph beginning on p. 4 and updated the Supplementary Discussion with additional details from the DFT calculation.

We note that Vitali et al. do perform ab initio calculations of isolated TbPc₂. However, it is well known that the generalised gradient approximation (GGA) exchange correlation functional used in the study of Vitali et al. does not properly describe strongly localized electronic states like f-states in rare earth metals. To account for this properly in the theoretical description of the strongly localized f-states requires the use of the GGA+U method, where a Hubbard U

describes the on-site Coulomb interactions. We have included this in our calculations, but Vitali et al. did not because they “do not aim at a precise ab initio characterization of the adsorption and electronic states of the complete system formed by TbPc₂ supported by a metal surface” since “this goal would be computationally too demanding”. Instead, their “goal is to give a qualitative picture of the physical properties of the supported TbPc₂ molecules through their electronic and magnetic characterization.” As a result, for example, the Tb f-states in Fig. 2 of their paper are close to Fermi level even though other STM studies [e.g. Nature Commun. 3, 953 (2012)] do not detect any f-states for TbPc₂ on Co/Ir(111) but rather only ligand states.

Furthermore, while Vitali et al. did explore the impact of adsorption on the molecule, they did not directly calculate the configuration of a molecule adsorbed on a Cu surface. Instead they studied a single molecule sandwiched between two Au(111) surfaces, which “closely corresponds to the situation of such a molecule in a break junction experiment, and that is of direct relevance for applications in molecular electronics”. Therefore, we have not included a direct comparison of their results with ours.

b) The shape of Kondo resonance

By analyzing the zero-bias peak with the Fano resonance function, the authors deduces $q \sim 1$ which suggest the tunneling electron is injected into both the Kondo resonance and the continuum. Actually the shape of the peak is similar to the Kondo resonance reported for a 3d magnetic metal atom deposited on Au(111) surface.

This q parameter suggests a strong coupling between the orbital, which is responsible for the Kondo resonance, and the substrate. It is quite reasonable for the case of 3d magnetic metal atom case. However, the 4f atom of DyPc₂ is not directly attached to the surface. In addition, even the Kondo resonance is caused by its hybridization with d orbital of the Dy atom, the referee cannot consider a mechanism which makes the coupling with the substrate much stronger than that of the TbPc₂, the latter of which shows a q corresponding to a weak coupling with the substrate.

The authors should provide more detail analysis for the origin of the parameter q to be ~ 1 .

c) Related to issue b). The Dy atom is, in a sense, shielded by the top Pc ligand. The STM tip cannot access physically to the Dy atom, and it is assumed that the tip position is regulated largely by the DOS of the Pc ligand and not by the Dy orbitals. The tunneling electrons are injected first into the orbital of the ligand. This is totally different from the case of 3d magnetic atom on non-magnetic metal surface case. The authors should discuss how the Kondo resonance is probed with such a tunneling path.

For DyPc₂ on Cu(001), the lower Pc ligand strongly interacts with the copper surface such that its electronic states form broad metallic-like bands (as shown in Supplementary Fig. 2). Since the df-atomic hybrids of Dy hybridise with some of the orbital states of the Pc ligands (as shown in Supplementary Fig. 3), this suggests a strong coupling between the orbitals responsible for the Kondo resonance and the substrate. Therefore, the electrons are injected in the upper Pc ligand and then tunnel through a hybrid molecule-surface state with weight on the upper Pc ligand, possibly the Dy df atomic hybrid, and the lower Pc ligand/copper substrate hybrid. The value of $q \sim 1$ observed in our experiments suggests that a roughly equal number of tunnelling paths involve ligand states that do and do not incorporate the Dy df atomic hybrid.

To explain this in more detail, we have added text to the first paragraph on p. 5.

d) comparison with bare Ln atom adsorbate

Although the authors discuss the difficulty of the detection of 4f induced Kondo resonance, there is a recent report like following. More updated description for the detection of 4f Kondo resonance should be provided.

'Absence of a spin-signature from a single Ho adatom as probed by spin-sensitive tunneling' by Steinbrecher et al. DOI: 10.1038/ncomms10454

We thank the referee for highlighting this recent high quality and timely work, and have added a reference to it accordingly.

e) Flexible configuration change of molecule; asymmetric electronic structure;
Asymmetric electronic configuration is discussed as a mechanism of the hybridization of 4f orbital with other orbitals.

However, the conformation change of the molecule can be seen for many molecule with adsorption on the surface. The authors should provide the evidence that the asymmetric STM image is not induced from the conformation change of the molecule upon adsorption.

If the asymmetry observed were the result of a conformation change in the molecule, most likely driven by the electric field in the STM junction, then we would expect the molecule to be asymmetric when imaged at both positive and negative bias voltages. However, we observe that the molecule is symmetric when imaging at negative bias (Fig. 3), and that the asymmetry is not strongly affected by the height of the tip, which can be varied by imaging the molecule using a different setpoint current. This suggests that the asymmetry is not the result of a physical deformation of the upper Pc ligand of the molecule but rather arises because of electronic interactions with the substrate. To explain this more clearly, we have added text to the last paragraph on p. 6.

Reviewer #3 (Remarks to the Author):

The authors present evidence of a Kondo resonance arising from the coupling between the f-states of a single-molecular magnet and the metallic states of the supporting surface. The work seems to me very carefully performed, with high scientific rigour, well inserted into the current literature, and, quite importantly, reported in a way that clearly emphasises the new physics displayed by the specific system under analysis. In my opinion this work is a valuable addition to the scientific literature of surface-supported single-molecular magnets, and would recommend publication after the authors considered the following recommendations.

The work reports two main conclusions: a) the closely confined magnetic moment carried by the 4f states can be accessed through its electronic coupling with the metallic substrate; and b) this coupling can be controlled at the molecular scale by varying the symmetry and orientation of the interacting molecule. I find that the first conclusion is very well documented, reported, and supported by clear figures. The second conclusion is in my opinion not equally well explained. Its description is limited to the last two paragraphs of the result section, that are in my opinion not fully convincing and should be expanded so as to better support conclusion b. Incidentally, the role of van der Waals interactions into this effect, i.e. change the relative rotation of one Pc ligand with respect to the other, is not evident nor obvious to me.

In this study we report on how the strength of the Kondo resonance can be spatially modulated by asymmetric local *intramolecular* variations in the coupling between the DyPc₂ molecule and the substrate. This asymmetry arises because the lower Pc ligand is at an angle from the symmetry axes of the Cu(001) surface. We note that the two orientations of the DyPc₂ on Cu(001) shown in Figs. 1a and 3a are mirror images, and therefore are by symmetry equivalent. To further clarify the arguments that this asymmetry arises from electronic rather than conformational changes, we have modified the text in the last paragraph on p. 2 and the last few paragraphs on p. 7.

In the future, it may be possible to modify the orientation of the molecule, and therefore control the nature of the coupling to the substrate. This could be done, for example, by chemically functionalising the DyPc₂ molecule (for instance with fluorinated or chlorinated Pc ligands). However, such an investigation is beyond the scope of the present study.

To help understand the origin of the asymmetry, we note that the isolated molecule has ligands that are 45° apart, as expected from the symmetry of the system. However, when the molecule is adsorbed on the surface, the ligands are rotated by a few additional degrees into an asymmetric configuration, which can only arise through an interaction between the top ligand and the surface. Since the upper ring is too far (~6 Å) above the metal surface to interact with it directly, we suggest that in this case the van der Waals interactions should be important to stabilise the geometry of the adsorbed molecule. Unfortunately, there is no simple way to determine more directly the impact of the van der Waals interactions in this system from our calculations. To clarify these points, we have revised our discussion on p. 7.

The authors support and explain the experimental STS measurements by high-quality DFT calculations. These reveals the hybridisation of the Dy-f states with the on-site s and d states. This conclusion is at the basis of the interpretation of the results. My question is on the specific charge population analysis used to partition the charge density and to obtain that result. This partition is non unique and there are several possibilities to do so. Projecting the

charge in within a sphere of a given radius on the Dy atomic orbitals is the method used in this work. How much the reported hybridisation arises from the ligand charge density entering into the projection sphere? How would the result change as a function of the sphere radius or by employing a different charge population analysis? The authors could strengthen their conclusion by demonstrating that the reported hybridisation is independent on the chosen charge analysis.

Our analysis of the onsite hybridization of the Dy-f states with the on-site s and d states for both the isolated molecule and with the molecule on the surface is based on projector augmented wave method (PAW), which provides an all electron description for the valence electrons in a given atomic sphere. Note that the radius of the atomic sphere is chosen during the generation of the PAW pseudopotential and kept fixed in all subsequent calculations. In particular, in our study we employed hard PAW pseudopotentials with atomic radii as small as possible to accurately describe the molecule-surface structural relaxation. Therefore, in the PAW formalism these small atomic radii set the lowest limit of the amount of charge density with s, p, d, f atomic-like character as compared to PAW pseudopotentials with larger atomic radii. The use of softer PAW pseudopotentials with larger atomic radii could only increase the electron counts found within the Dy atomic integration sphere and hence further reduce our estimate of the charge transfer to the ligands, compared to the hard PAW pseudopotentials employed in our calculations.

To explain this more clearly, we have added text to the Supplementary Discussion section.

REVIEWERS' COMMENTS:

Reviewer #1 (Remarks to the Author):

The authors have satisfactorily answered all the points raised by this referee and publication of this manuscript in Nature Communications journal is recommended.

Reviewer #2 (Remarks to the Author):

The authors revised and added phrases to the issues raised by the referee. The revised manuscript should be published in Nat Comm as it is.

Reviewer #3 (Remarks to the Author):

The authors carefully addressed my previous recommendations and, in general, all reviewers issue, modifying the text accordingly and providing additional details and comments. The present manuscript is of high scientific quality and it is in my opinion suitable for publication in Nature Communications.